# Magnetic field compatible circuit quantum electrodynamics with graphene Josephson junctions

J.G. Kroll [1], W. Uilhoorn[1], K.L. van der Enden[1], D. de Jong[1], K. Watanabe [2], T. Taniguchi[2], S. Goswami[1], M.C. Cassidy[1] & L.P. Kouwenhoven[1,3]

Circuit quantum electrodynamics has proven to be a powerful tool to probe mesoscopic effects in hybrid systems and is used in several quantum computing (QC) proposals that require a transmon qubit able to operate in strong magnetic fields. To address this we integrate monolayer graphene Josephson junctions into microwave frequency super-conducting circuits to create graphene based transmons. Using dispersive microwave spectroscopy we resolve graphene's characteristic band dispersion and observe coherent electronic interference effects confirming the ballistic nature of our graphene Josephson junctions. We show that the monoatomic thickness of graphene renders the device insen-sitive to an applied magnetic field, allowing us to perform energy level spectroscopy of the circuit in a parallel magnetic field of 1 T, an order of magnitude higher than previous studies. These results establish graphene based superconducting circuits as a promising platform for QC and the study of mesoscopic quantum effects that appear in strong magnetic fields.

[1] QuTech and Kavli Institute for Nanoscience, Delft University of Technology, 2600 GA Delft, The Netherlands. [2] Advanced Materials Laboratory, National Institute for Materials Science, 1-1 Namiki, Tsukuba 305-0044, Japan. [3] Microsoft Station Q Delft, 2600 GA Delft, The Netherlands. Correspondence and requests for materials should be addressed to L.P.K. (email: Leo.Kouwenhoven@Microsoft.com)

A superconducting transmon qubit[1] resilient to strong magnetic fields is an important component for proposed topological[2–4] and hybrid quantum computing (QC) schemes[5,6]. A transmon qubit consists of a Josephson junction (JJ) shunted by a large capacitance, coupled to a high quality factor superconducting resonator. In conventional transmon devices, the resonator is fabricated from Al and the JJ is fabricated from an Al/AlO$_x$/Al tunnel junction[1], both of which cease operation above the critical magnetic field of bulk Al, ~10 mT. Even when considering alternative type II superconductors such as NbTiN or MoRe that can sustain superconductivity beyond $B = 8$ T[7], when subjected to a strong magnetic field the superconductor will experience detrimental effects such as reduction of the superconducting gap, increased quasiparticle generation[8] and the formation of Abrikosov vortices that cause resistive losses in a microwave field. In addition to disrupting the superconductivity, magnetic flux penetrating the JJ produces electron interference effects that reduce the Josephson energy $E_J$ and strongly suppress the transmon energy spectrum. If the transmon is to be used for fast quantum gates, fast charge-parity detection and long range quantum state transfer in QC schemes[3,9,10] we are compelled to consider alternatives to conventional Al based JJs. Proximitised semiconducting nanowires, acting as gate-tuneable superconductor-normal-superconductor JJs[11] have been used successfully in a variety of microwave frequency superconducting circuits, allowing for studies of Andreev bound states[12,13], electrically tuneable transmon qubits[14,15] and transmons that exhibit substantial field compatibility[16]. Graphene JJs are an attractive alternative as they exhibit ballistic transport, high critical currents[7,17,18] and the atomic thickness of the graphene junction greatly reduces flux penetration, protecting the JJ from orbital interference effects that would suppress $E_J$ in high parallel fields. When combined with geometric techniques that protect the superconducting film, such as critical field enhancement[19] and lithographically defined vortex pinning sites[20,21], the transmon circuit can be protected at magnetic fields relevant to these proposals, which approach and in some cases exceed 1 T[22–24].

In this work we report the integration of ballistic graphene JJs into microwave frequency superconducting circuits to create graphene based transmons. Using dispersive microwave spectroscopy we resolve the characteristic band dispersion of graphene, and observe coherent electronic interference effects that confirm the ballistic nature of our graphene JJs. We perform energy level spectroscopy at $B_\| = 0$ T to resolve a linewidth of $\simeq 400$ MHz. Although the large linewidths prevent coherent qubit control, we demonstrate the device is insensitive to the applied magnetic field up to $B_\| = 1$ T.

## Results

**Device structure**. Figure 1a shows an optical microscope image of a typical graphene transmon device. It consists of four $\lambda/4$ coplanar waveguide (CPW) resonators multiplexed to a common feedline. Each resonator is capacitively coupled to a graphene transmon, with the graphene JJ being shunted by capacitor plates that provide a charging energy $E_C \simeq 360$ MHz. The resonators and capacitor plates are fabricated from 20 nm NbTiN due to its enhanced critical magnetic field[19], and we pattern the resonators with a lattice of artificial pinning sites to protect the resonator from resistive losses due to Abrikosov vortices[20,21]. The van der Waals pickup method is used to encapsulate monolayer graphene (G) between two hexagonal boron nitride (hBN) flakes and deposit it between the pre-fabricated capacitors plates (Fig. 1b)[7], before contacting the hBN/G/hBN stack by dry etching and sputtering MoRe. In this work, we present results from two graphene JJ transmon devices, with slightly different fabrication

techniques. Device A uses a Ti/Au gate stack deposited directly on the hBN, before the junction is shaped via dry etching. Device B is shaped (Fig. 1c) before a Ti/Au gate stack with a SiN$_x$ interlayer is deposited (Fig. 1d).

**Dispersive Fabry–Perot oscillations**. We begin by performing spectroscopy of the resonator in device A as a function of the input power $P_{in}$ (Fig. 2a). Varying the resonator's photon occupation from $\langle n_{ph} \rangle \simeq 1000$ to $\langle n_{ph} \rangle = 1$ we observe a dispersive shift $\chi = f_r - f_{bare}$ in the resonator frequency $f_r$ from the high power value $f_{bare}$. This occurs due to a Jaynes-Cummings type interaction between the harmonic readout resonator and the anharmonic transmon spectrum, with the anharmonicity provided by the Josephson junction[25]. The magnitude of the shift $\chi = g^2/\Delta$ depends on the transmon-resonator coupling $g$, and the difference $\Delta = f_r - f_t$ between $f_r$ and the ground state to first excited state transition frequency $f_t = E_t/h \simeq \sqrt{8E_J E_C}/h$, allowing us to infer $E_J$ from $\chi$[1]. Studying $\chi$ as a function of gate voltage $V_G$ reveals the characteristic band dispersion of graphene (Fig. 2b) and allows the voltage at the charge neutrality point (CNP) $V_{CNP}$ to be identified. At negative $V_G - V_{CNP}$, the chemical potential $\mu$ is below the CNP and the graphene is in the p-regime where holes are the dominant charge carrier. Deep into the p-regime, the high carrier density ($n_C$) gives a large $E_J$, placing $f_t$ above the resonator and giving $\chi$ a small negative value (Fig. 2c). As $V_G$ approaches the CNP, the Dirac dispersion minimises the density of states reducing $E_J$ and $f_t$ to a minimum. Since $\chi = g^2/\Delta$, as $\Delta$ approaches zero, $\chi$ diverges. Once on resonance, the resonator acquires some characteristic of the qubit, significantly broadening the lineshape. Simultaneously, the critical photon number $n_{Crit} = \Delta^2/4g^2$ collapses[26], moving the measurement into the 'transitionary' regime between high and low photon number as in Fig. 2a, causing the anomalous lineshapes visible in Fig. 2c near CNP. As $V_G$ is increased past the CNP, $n_{Crit}$ and the lineshapes recover, with electrons becoming the dominant charge carrier and $E_J$ increasing to a maximum as expected from removal of the n-p-n junction formed by the contacts[7]. The p-regime also experiences periodic fluctuations in $E_J$ as a function of $V_G$ due to coherent electron interference effects in a Fabry–Perot cavity formed by n-p interfaces at the MoRe contacts[7]. Extracting a line

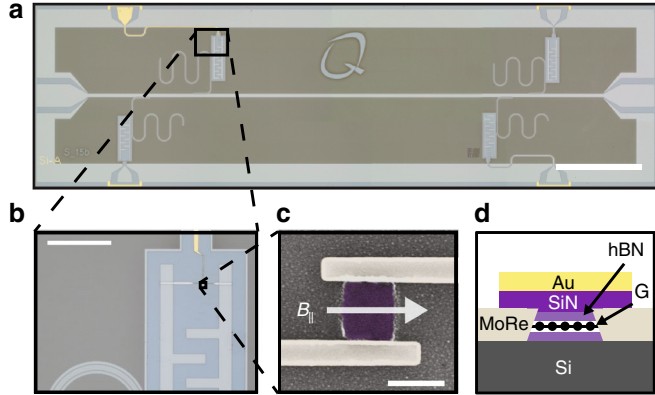

**Fig. 1** Device structure **a** Optical image showing multiple CPW resonators frequency multiplexed to a common feedline (device B). Scale bar 1 mm. **b** Zoomed optical image of the capacitor plates that shunt the Josephson junction, with the gate, junction and contacts visible. Scale bar 100 µm. **c** SEM micrograph of a contacted boron nitride-graphene-boron nitride stack before fabrication of the gate. A magnetic field $B_\|$ can be applied parallel to the film along the length of the junction contacts using a 3-axis vector magnet. Scale bar 500 nm. **d** Cross sectional diagram showing the fully contacted and gated stack

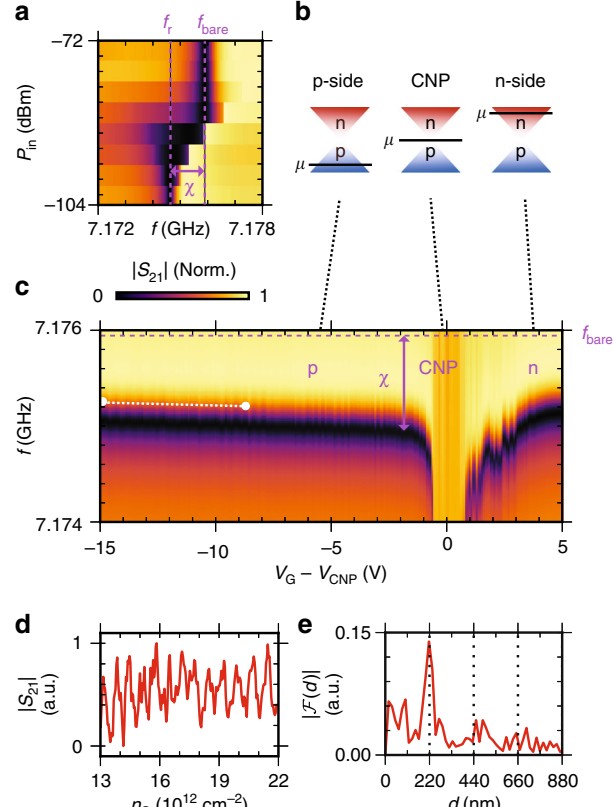

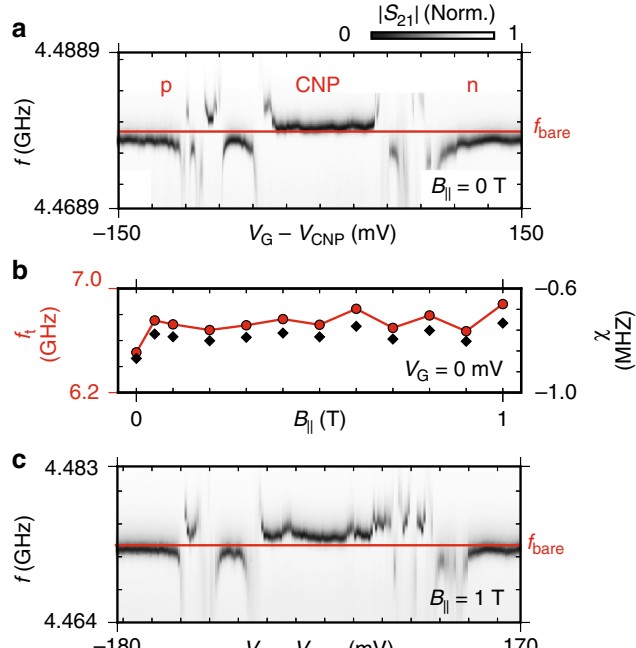

**Fig. 3** Dispersive shift as a function of $V_G$ and $B_\parallel$ **a** At $B_\parallel = 0$ T, $|S_{21}|$ (Norm.) versus $f$ and $V_G$ (with $V_{CNP} = 300$ mV subtracted) shows the symmetric band dispersion of graphene with additional fluctuations we attribute to UCF. **b** $f_t$ (red circles) extracted from $\chi$ (black diamonds) versus $B_\parallel$ at $V_G = 0$ V, showing $f_t$ is not significantly affected. **c** Repeating **a** at $B_\parallel = 1$ T with $V_{CNP} = 430$ mV subtracted confirms the graphene JJ behaves equivalently to $B_\parallel = 0$ T. The variation observable in **b** and shift in $V_{CNP}$ between **a** and **c** we attribute to slow gate drift

**Fig. 2** Resonator spectroscopy as a function of $P_{in}$ and $V_G$ **a** $|S_{21}|$ (Norm.) as a function of input frequency $f$ and input power $P_{in}$. At single photon occupancy the resonator experiences a frequency shift $\chi$ due to repulsion from an energy level above the resonator (device A). **b** Diagram of the Dirac cone band structure of graphene. Changing $V_G$ to tune $\mu$ allows the dominant charge carriers to be varied between hole, charge neutral and electron-like regimes. **c** At single photon occupancy, $|S_{21}|$ (Norm.) is measured as $f$ and $V_G$ are varied, with the voltage at CNP ($V_{CNP} = 7.8$ V) subtracted. In the p-regime, $\chi$ oscillates as $V_G$ is varied. We extract the charge carrier density $n_c$ **d** from the white linecut to generate a Fourier transform **e** that is consistent with Fabry-Perot oscillations in a cavity of $d = 220$ nm

trace (white line Fig. 2c) to study the modulation in $|S_{21}|$ with $n_C$ (Fig. 2d), and performing a Fourier transform (Fig. 2e) gives a cavity length of 220 nm in agreement with the device dimensions. The observation of a Dirac dispersion relation in combination with coherent electron interference effects confirm the successful integration of ballistic graphene JJs into a superconducting circuit.

**Insensitivity to applied parallel magnetic field**. In device B we observe additional coherent electronic interference effects in the form of universal conductance fluctuations (UCF)[14,27]. As we move from the p to the CNP regime, $\chi$ is seen to diverge repeatedly as $f_t$ anti-crosses multiple times with $f_r$ (Fig. 3a). This behaviour is repeated moving from the CNP to the n-regime, where $E_J$ is again maximised. We demonstrate the field compatibility of the junction by applying a magnetic field $B_\parallel$ along the length of the junction contacts, parallel to the plane of the film, using the resonator as a sensor for field alignment (see Supplementary Figs. 1 and 2 for alignment procedure details). Monitoring $\chi$ as $B_\parallel$ is varied between 0 and 1 T (Fig. 3b) and calculating $f_t$ (using $g = 43$ MHz, extracted from measurements in Fig. 4), demonstrates that $\chi$ and thus $E_J$ are not significantly

affected by the applied $B_\parallel$. The small amount of variation observed is attributed to charge noise induced gate drift which was observed throughout the duration of the experiment. Studying $\chi$ as a function of $V_G$ at $B_\parallel = 1$ T (Fig. 3c) again reveals the characteristic Dirac dispersion as seen in Fig. 3a, with modified UCF and shifted $V_{CNP}$ due to slow gate drift. The insensitivity of $f_t$ to applied field and similarity of device operation at $B_\parallel = 0$ and 1 T confirm the field resilience of both the graphene JJ and superconducting circuit.

**Two tone spectroscopy in high parallel magnetic fields**. In order to better understand the microwave excitation spectra of our system we proceed to measure it directly via two-tone spectroscopy[1]. The readout tone is set to $f_r$ whilst a second tone $f_d$ is used to drive the circuit. Excitation of the system results in a state dependent shift of the resonator frequency, and is detected by measuring the change in the complex transmission $S_{21}$ at $f_r$. At $V_G = 0$ V (p-regime), two-tone spectroscopy at $B_\parallel = 0$ and 1 T (Fig. 4a) can be fitted with a Lorentzian to extract the transmon transition $f_t \simeq 5.2$ GHz and transition linewidth $\gamma \simeq 400$ MHz. At $B_\parallel = 1$ T, $f_t$ and thus $E_J$ differ only slightly with $\gamma$ increasing slightly from 350 to 425 MHz. The transmon resonator coupling $g = \sqrt{\chi\Delta} = 43$ MHz is extracted from the observed dispersive shift $\chi$ and detuning $\Delta$, and used in the calculation of $f_t$ in Fig. 3. We attribute the change in $f_t$ from Fig. 3b and the large $\gamma$ to the dielectric induced charge noise mentioned previously. An estimate of $E_J = 40.2$ μeV $= 9.72$ GHz can be provided using the relation $E_t = hf_t \simeq \sqrt{8E_J E_C}$. Performing two-tone spectroscopy in the n-regime while tuning $V_G$ reveals a gate-tunable energy level that is visible above and below the resonator (Fig. 4b, $V_{CNP}$

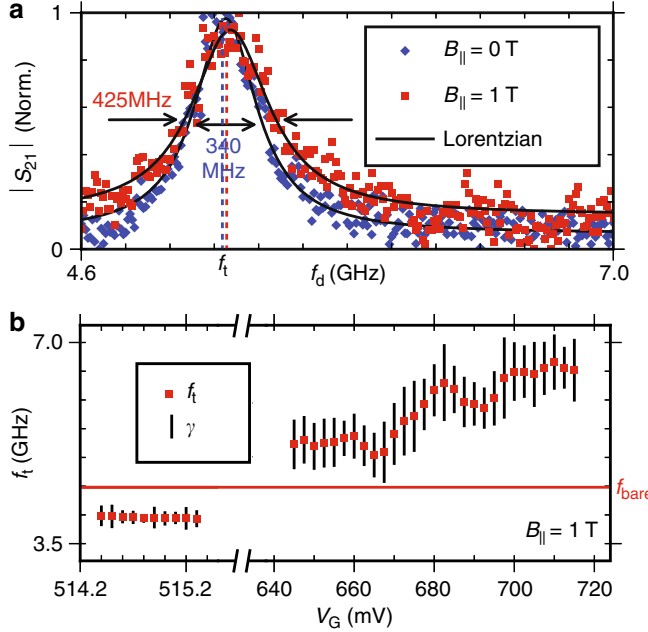

**Fig. 4** Two tone spectroscopy **a** Normalised $|S_{21}|$ at $f_r$ as $f_d$ is varied can be fitted to extract $f_t$ and $\gamma$ at $V_G = 0$ V. At $B_{||} = 1$ T, $\gamma$ shows a 25% increase compared to $B_{||} = 0$ T. **b** At $B_{||} = 1$ T, $f_t$ and $\gamma$ are extracted as $V_G$ is varied, demonstrating $f_t$ can be swept over a wide frequency range. Lines bisecting each $f_t$ are not error bars, but represent the extracted $\gamma$ at each $f_t$

not specified due to gate drift during measurement) that can be fitted to extract $f_t$ and $\gamma$, giving a minimum linewidth of 166 MHz (see Supplementary Fig. 3 for the raw data).

## Discussion

The observation of a transition and the inferred high value of $E_J$ in the n and p-regimes (Fig. 4a) provides additional confirmation of the electron-hole symmetry expected in graphene. Additional measurement of the higher order two-photon $f_{02}$ transition would allow for exact measurements of $E_J$ and $E_C$ via diagonalisation of the Hamiltonian, enabling investigations into mesoscopic effects of interest in graphene JJs[28,29]. Importantly, the transition and thus $E_J$ can be varied over a wide frequency range, satisfying a key requirement for implementation into topological QC proposals[3]. If graphene based transmons are to be successfully implemented into these proposals however, the large linewidths that currently prevent measurements of relaxation and coherence lifetimes $\left(T_1, T_2^*\right)$ must be reduced. We believe that material improvements to the dielectric materials can achieve this.

In conclusion, we have integrated a graphene JJ into a superconducting circuit to make a graphene based transmon. Additionally, we have achieved operation at $B_{||} = 1$ T, a magnetic field more than an order of magnitude higher than previous studies[16,30]. While the broad linewidths prevented the demonstration of coherent qubit control, these results establish graphene based microwave circuits as a promising tool for topological and hybrid QC schemes, and for probing mesoscopic phenomena of interest at high magnetic fields.

## Methods

**Sample fabrication**. To fabricate the two devices (A and B) 20 nm of NbTiN is sputtered onto intrinsic Si wafers in an Ar/N atmosphere. The resonators, feedline and transmon are reactive ion etched in an $SF_6/O_2$ atmosphere. In this etching step, an array of artificial pinning sites is also defined. Monolayer graphene is

encapsulated between two hBN flakes ($t \simeq 15$ nm each), then deposited between pre-fabricated capacitors using a PMMA based van der Waals pickup method. Contact to the graphene stack is made by etching in a $CHF_3/O_2$ environment, followed by sputtering MoRe ($t = 80$ nm). As shown in Supplementary Fig. 4, device A was contacted to give a junction length of 300 nm. A Ti/Au top gate is then sputtered on top of the stack. The device is then shaped in a $CHF_3/O_2$ plasma to be $1000 \times 300$ nm$^2$ in size. Device B was contacted to provide a junction length of 500 nm. The long thin leads were geometrically restricted in two dimensions, making it less favourable for vortices to form, protecting the superconductivity of the contacts proximitising the junction. The junction is then shaped in a $CHF_3/O_2$ plasma to be $500 \times 500$ nm$^2$. A $SiN_x$/Ti/Au top gate stack is then sputtered to give full junction coverage, giving greater control of $\mu$ in the junction.

**Sample characterisation**. All measurements were performed in a dilution refrigerator with a base temperature of 15 mK. The samples were enclosed in a light tight copper box, and thermally anchored to the mixing chamber. An external magnetic field is applied to the sample using a 3-axis vector magnet. The two different measurement configurations used in this manuscript are depicted in Supplementary Fig. 5. Two coaxial lines and one DC line were used to control the sample. The sample was connected to the DC voltage source by a line that was thermally anchored at each stage and heavily filtered at the mixing chamber by low frequency RC, $\pi$ and copper powder filters. The line used to drive the feedline input was heavily attenuated to reduce noise and thermal excitation of the cavity, allowing the single photon cavity occupancy to be reached. The output line of the feedline was connected to an isolator (Quinstar QCI-080090XM00) and circulator (Quinstar QCY-060400CM00) in series to shield the sample from thermal radiation from the HEMT amplifier (Low Noise Factory LNF-LNC4-8_C) on the 4 K stage. Resonator spectroscopy of device A was performed using circuit (a) to measure the amplitude and phase response of the complex transmission $S_{21}$ as the frequency was varied. Resonator and two-tone spectroscopy of device B was performed using circuit (b), with a splitter used to combine the readout and excitation tones. This allows the complex $S_{21}$ to be measured, but only at fixed resonator readout frequency otherwise only $|S_{21}|$ can be recorded.

## Data availability

The data used to support this study, and the code used to generate the figures are available from a public data repository here https://doi.org/10.4121/uuid: b7340d11-e47e-44eb-a60d-679d758c7160. (ref. [31]).

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

## Acknowledgements

We thank D.J. van Woerkom for fabrication assistance, M.W.A. de Moor and A. Proutski for helpful discussion and L. DiCarlo, C. Dickel and F. Lüthi for experimental advice and software support. This work was supported by the European Research Council (ERC), The Dutch Organisation for Scientific Research (NWO), and Microsoft Corporation Station Q.

## Author contributions

K.W. and T.T. grew the hBN crystals, J.G.K. and W.U. fabricated the devices, J.G.K., K.L. v.d.E. and D.d.J performed the measurements and J.G.K. and K.L.v.d.E. analysed the measurements. The manuscript was prepared by J.G.K. with K.L.v.d.E., S.G., M.C.C. and L.P.K. providing input. S.G., M.C.C. and L.P.K. supervised the project.

## Additional information

**Competing interests:** The authors declare no competing interests.

