## [Peer Review File · Nature Communications]

Reviewers' comments:

Reviewer #1 (Remarks to the Author):

In this article, the authors describe the fabrication and measurements of graphene Josephson junctions coupled to a superconducting resonator, as transmons. These measurements at microwave frequencies are conducted at zero magnetic field, and at 1 Tesla. Varying the gate voltage allows the excitation energy from the ground to first excited states to be tuned.

The paper is clearly written, with clear figures and figure captions. The devices themselves are well-designed, and the results may be of interest to researchers seeking to incorporate Josephson junctions in an environment with strong magnetic fields. As a result, I am in favor of publishing this article.

However, I have a number of questions/comments that will hopefully improve this article prior to publication.

- A significant justification for the measurements presented in this paper appears to be "the transmon can be protected at magnetic fields relevant to these proposals, which in some cases exceeds 1T" (lines 68-69). In perusing the references, I found some applications using 100 mT or 500 mT. It would be helpful if the authors were more explicit in noting/citing which proposals they were referring to.
- The abstract and early sections of the paper emphasize quantum computation applications. However, in the conclusion, it is noted, "While the broad linewidths prevented the demonstration of coherent qubit control..." This should be made clear earlier in the paper.
- The gate drift (suggested in the caption of Figure 3) was very substantial relative to the scale of the figures. So, it may be helpful if all figures plotting V_G were instead $V_G - V_{CNP}$ (as in Figure 2c, rather than figures 3a, 3c, and 4b). If this is not possible, some mention of the charge neutrality point should be indicated in either the figure or caption (which is lacking in Figure 4b).
- Lines 185-189 "Performing two-tone spectroscopy whilst tuning V_G reveals a gate-tunable energy level that is visible on the p-side and n-side of the CNP, reiterating the electron-hole symmetry expected in graphene." This sentence appears to refer to Figure 4b. However, there is no indication of the charge neutrality point on this figure, nor are there any obvious symmetries of the figure (especially because the x-axis is not linear). This section and/or the figure should be made more clear.
- The paper would be strengthened by adding citations for two-tone spectroscopy (on line 99) and universal conductance fluctuations (line 141).
- In section V of the supplemental material, "it was found that the orientation of the field with respect to the leads was of key importance." In reviewing Figure 1 of the main text, it appears that the junction contacts were parallel to the common feedline of device B. It is unclear whether or not it was also parallel for device A. If it was, have the authors considered whether the effect of varying the orientation of the in-plane magnetic field may have been due to interactions with the stripline rather than the leads?
- Did the authors find any evidence of gate drift beyond results similar to Figures 3a and 3c? Are there any indications of its cause? If not, using V_G for tuning purposes may be problematic.

Finally, some relatively minor issues:

- Reference 18 is missing essential information.
- Supplemental material section II includes "...and heavily filtered at the mixing chamber by low RC, n and copper powder filters." I presume that was intended to read "low-temperature RC...".

Reviewer #2 (Remarks to the Author):

The manuscript "A graphene transmon operating at 1 T" by J. G. Kroll et al. describes a transmon structure with graphene Josephson junction and its spectroscopic properties with respect to electrostatic gating conditions and in-plane magnetic field. The authors resolve the qubit transition via two-tone spectroscopy. Although its linewidth is too large to show coherent qubit control, authors confirm the linewidth is resilient to magnetic field up to 1 Tesla, which suggest that graphene-based transmon structure can be used for studying topological qubits under magnetic field. It is an important and meaningful result toward (topological) superconducting qubits operations under intense magnetic fields. I suggest its publication in Nature Communications with minor revisions.

1. Current title is misleading. Any coherent qubit operations such as Rabi flops have not been demonstrated in this work. However, the current title reads like the authors demonstrated an operational transmon qubit at 1 T. I suggest the authors to update the title to present the nature of their work more accurately.
2. Why does the linewidth abruptly increase then decrease as power gets lowered near the critical power in Fig. 2a?
3. Clarify V_{CNP} for device A and B in the manuscript. Where is V_{CNP} for Fig. 4b?
4. There are features that look like anti-crossings between the cavity and qubit levels in Fig. 3, but it is difficult to resolve visually. It would be more convincing if the authors show zoomed-in anti-crossing areas.
5. Near CNP, why does the dip in $|S_{21}|$ disappear in Fig. 2c, but show up in Fig. 3a? Why dip stays stable at CNP in Fig. 3a, but not in Fig. 2c.
6. In Figs. 3a,c and 4a, the authors indicate observed charge noise and its possible effect on the qubit linewidth. How much charge noise is observed on average? How much does qubit linewidth increase due to the observed amount of charge noise? The discussion about the way of improving linewidth for coherent quantum operation would strengthen the manuscript.
7. I think E_J in GHz is more helpful for the readers.
8. At even higher magnetic fields above 1 T, the qubit transition should be affected and eventually disappear. Either a plot of γ vs. B , or the qubit transition spectrum with significantly broadened linewidth at certain field strength, would make the conclusion more convincing. If those data are not available, how about to show data at slightly misaligned magnetic field in the manuscript or in the supplementary information?

Reviewer #3:

authors: $|S_{21}|$ (Norm.) versus f and V_G shows the band dispersion of graphene with additional fluctuations we attribute to UCF.

How does the linear dispersion of graphene show up in this diagram? It is unclear what the authors mean.

Reviewer #1:

1. A significant justification for the measurements presented in this paper appears to be "the transmon can be protected at magnetic fields relevant to these proposals, which in some cases exceeds 1T" (lines 68-69). In perusing the references, I found some applications using 100 mT or 500 mT. It would be helpful if the authors were more explicit in noting/citing which proposals they were referring to.

The cited proposals for topological quantum computation require a topological state to be induced in the semiconducting systems used to create the Majorana bound states. Although the magnetic field required to do this depends upon many factors, current research suggests that in the leading material systems (InAs 2DEGs, as well as InSb and InAs nanowires) this typically occurs at, or in excess of, 1T. As suggested by the referee, we have cited sources that demonstrate that the topological regime is reached at or beyond 1T for these material systems.

2. The abstract and early sections of the paper emphasize quantum computation applications. However, in the conclusion, it is noted, "While the broad linewidths prevented the demonstration of coherent qubit control..." This should be made clear earlier in the paper.

We agree with the Reviewer, and in order to prevent confusion have clarified this early in the paper.

3. The gate drift (suggested in the caption of Figure 3) was very substantial relative to the scale of the figures. So, it may be helpful if all figures plotting V_G were instead $V_G - V_{\text{CNP}}$ (as in Figure 2c, rather than figures 3a, 3c, and 4b). If this is not possible, some mention of the charge neutrality point should be indicated in either the figure or caption (which is lacking in Figure 4b).

Plotting the figures as $V_G - V_{\text{CNP}}$ was originally avoided out of concern that it would confuse the reader and obscure the fact that the CNP was varying due to gate drift.

We have replotted the figures and made explicit that V_{CNP} is different for each panel. We have not done this in the case of Fig4a, as we are unable to accurately determine the value of V_{CNP} during the measurement. Measurements before and after show that it changed significantly during the measurement however, confirming that we cannot assume the previously measured values.

4. Lines 185-189 "Performing two-tone spectroscopy whilst tuning V_G reveals a gate-tunable energy level that is visible on the p-side and n-side of the CNP, reiterating the electron-hole symmetry expected in graphene." This sentence appears to refer to Figure 4b. However, there is no indication of the charge neutrality point on this figure, nor are there any obvious symmetries of the figure (especially because the x-axis is not linear). This section and/or the figure should be made more clear.

The reviewers confusion stems from the fact that this sentence was ambiguously written. The confusion stems from the fact that:

1. Measurements before and after the measurements of Fig 4b indicate that V_{CNP} drifted significantly during this long time frame. As such, we cannot confidently state a value of V_{CNP} , hence why it was omitted from the figure.
2. With regards to 'reiterating electron-hole symmetry expected in graphene', this refers to the fact that two-tone spectroscopy was successfully performed on the n-side (Fig 4b) and the p-side (a single measurement shown in Fig4a at $V_G = 0V$).

To remedy this:

1. A note in the text explaining why V_{CNP} could not be measured has been added.
2. The manuscript has been edited to clarify this.

5. The paper would be strengthened by adding citations for two-tone spectroscopy (on line 99) and universal conductance fluctuations (line 141).

We agree with the reviewer, and have done so.

In section V of the supplemental material, "it was found that the orientation of the field with respect to the leads was of key importance." In reviewing Figure 1 of the main text, it appears that the junction contacts were parallel to the common feedline of device B. It is unclear whether or not it was also parallel for device A. If it was, have the authors considered whether the effect of varying the orientation of the in-plane magnetic field may have been due to interactions with the stripline rather than the leads?

In device A, it was also parallel, however no magnetic field studies were performed on device A. A significant amount of unpublished work involving magnetic field studies on resonators multiplexed to a common feedline make us confident that the observed behavior was not due to interactions of the magnetic field and the stripline.

6. Did the authors find any evidence of gate drift beyond results similar to Figures 3a and 3c? Are there any indications of its cause? If not, using V_G for tuning purposes may be problematic.

Gate drift was visible throughout the measurement, as repeated measurements of the same gate range would regularly produce shifted features, with the magnitude of the shift depending upon the length of time that had elapsed, but not in any monotonic or predictable way. Such gate drifts were not visible in device A (Fig. 2), which suggests that the sputtered SiN_x that serves as a top gate for device B could be the source of the drift. This is in agreement with gate drift observed in other

devices with sputtered SiN_x, measured in our other experiments.

Finally, some relatively minor issues:

7. Reference 18 is missing essential information.

This occurred due to a typesetting problem. This has been remedied in the latest version of the manuscript.

8. Supplemental material section II includes "...and heavily filtered at the mixing chamber by low RC, π and copper powder filters." I presume that was intended to read "low-temperature RC...".

This was meant to read as 'low frequency'. The error has been corrected.

Reviewer #2:

1. Current title is misleading. Any coherent qubit operations such as Rabi flops have not been demonstrated in this work. However, the current title reads like the authors demonstrated an operational transmon qubit at 1 T. I suggest the authors to update the title to present the nature of their work more accurately.

While writing the manuscript we were careful to not claim or imply that we had observed coherent qubit control. However, to make this more explicit, we have changed the title to: "Magnetic field compatible circuit quantum electrodynamics with graphene Josephson junctions".

2. Why does the linewidth abruptly increase then decrease as power gets lowered near the critical power in Fig. 2a?

The transitional regime between the 'high power' and 'low power' regimes is something that has been experimentally studied extensively but is currently not well understood, even in traditional SIS transmons. As such, we are unable to draw conclusions about the anomalous lineshapes and linewidth changing as the single photon regime is approached.

3. Clarify V_{CNP} for device A and B in the manuscript. Where is V_{CNP} for Fig. 4b?

As discussed previously in comment 4 for reviewer 1, the slow gate drift and long measurement time of several days means that we are unable to accurately determine V_{CNP} in this figure. A measurement of V_{CNP} several days later showed that it had shifted to 520 mV, suggesting that the drift throughout the measurement was likely significant.

4. There are features that look like anti-crossings between the cavity and qubit levels in Fig. 3, but it is difficult to resolve visually. It would be more convincing if the authors show zoomed-in anti-crossing areas.

We thank the reviewer for his comment, but we do not have a higher resolution dataset than that provided in the figure, and feel that providing a zoomed-in section for this section would interfere with the figures formatting without contributing scientifically.

5. Near CNP, why does the dip in $|S_{21}|$ disappear in Fig. 2c, but show up in Fig. 3a? Why dip stays stable at CNP in Fig. 3a, but not in Fig. 2c.

In Fig 2c, two separate effects combine to give the behavior observed.

1. This scan of f versus V_G was acquired at a constant readout power P_{in} . This power was calibrated to be below the critical photon number n_{crit} at $V_G = 0V$ to accurately resolve the dispersive shift χ . As the CNP is approached, the qubit approaches and starts to anticross with the resonator, reducing Δ to 0. This causes the critical photon number to collapse $n_{crit} = \Delta^2/(4g^2)$. As the readout power is constant throughout the measurement, the linescans near the CNP are now in the 'transitory' regime of a power scan (Fig 2a), making the line traces anomalous and difficult to interpret.
2. Simultaneously, as Δ is reduced to 0, the resonator and qubit more strongly couple and the resonator starts to acquire some of the characteristics of the qubit. Given the large linewidths measured in Fig 4, this would broaden the resonators lineshape to be larger than the measurement window, again causing an anomalous measurement.

To clarify this for the reader, we have added some discussion on these effects in the relevant section.

6. In Figs. 3a,c and 4a, the authors indicate observed charge noise and its possible effect on the qubit linewidth. How much charge noise is observed on average? How much does qubit linewidth increase due to the observed amount of charge noise? The discussion about the way of improving linewidth for coherent quantum operation would strengthen the manuscript.

Although quantitatively evaluating the charge noise may strengthen the manuscript, the broad linewidths measured mean that we are unable to demonstrate Rabi flops (as the referee correctly noted). Without coherent qubit control, we are unable to perform the dynamical decoupling measurements that would allow us to quantify the charge noise power spectrum. A comprehensive study on the effects of different gate dielectrics on the qubit linewidth would need to be performed to determine the best way (if any) to improve the qubit linewidth, which we feel lies outside the scope of the current manuscript.

7. I think E_J in GHz is more helpful for the readers.

We thank the reviewer for their comment, and although we think μeV are more useful for the DC transport community, we agree that GHz is preferable for the superconducting circuit community.

We have therefore added both now.

8. At even higher magnetic fields above 1 T, the qubit transition should be affected and eventually disappear. Either a plot of γ vs. B , or the qubit transition spectrum with significantly broadened linewidth at certain field strength, would make the conclusion more convincing. If those data are not available, how about to show data at slightly misaligned magnetic field in the manuscript or in the supplementary information?

Due to the orientation of our sample, we were unable to apply fields in excess of 1T. Studying the effect of a slightly misaligned magnetic field on the linewidth of the transmon transition would be of interest, but unfortunately our superconducting CPW resonators will not allow it. Although the resonators are engineered to sustain very high parallel fields, very small misalignments cause the

resonator frequency to fluctuate rapidly and the quality factor to become strongly suppressed, preventing its use as a sensor in two-tone spectroscopy. Evidence for this can be observed in the magnetic field alignment scans (Supplementary Figure 4) where misalignments of 0.01 of a degree are enough to severely affect the resonators performance. This would correspond to a perpendicular field of roughly 0.14 mT, far below the perpendicular magnetic fields that would start to affect the Josephson coupling in the junction.

Reviewer #3:

1. authors: “ $|S_{21}|$ (Norm.) versus f and V_G shows the band dispersion of graphene with additional fluctuations we attribute to UCF.” How does the linear dispersion of graphene show up in this diagram? It is unclear what the authors mean.

The specific feature of the graphene dispersion we are referring to is the existence of a charge neutrality point and the symmetry of the band dispersion around it, not the expected linear dispersion. **The text has been modified to clarify our intended point.**

REVIEWERS' COMMENTS:

Reviewer #1 (Remarks to the Author):

The authors have largely satisfied my concerns regarding the manuscript. As such, I recommend it for publication.

However, I note that Figure 3 could be improved by making the horizontal scales of (a) and (c) equal: one objective of these figures is to show "the similarity of device operation at $B_{||} = 0$ T and 1 T". It is reasonable to conclude that device operation is similar using the figures as they are, but equal scales will be helpful.

(If these are reformatted, a formatting quirk could be corrected at the same time: the text overlaying Fig. 3a has a white background that obscures some data - unlike Fig. 3c.)

Reviewer #2 (Remarks to the Author):

I thank authors for faithfully replying all raised questions. Now I would recommend this manuscript for the publication of Nature communications.

Response to Reviewers

Reviewer #1

The authors have largely satisfied my concerns regarding the manuscript. As such, I recommend it for publication.

However, I note that Figure 3 could be improved by making the horizontal scales of (a) and (c) equal: one objective of these figures is to show "the similarity of device operation at $B_{||} = 0$ T and 1 T". It is reasonable to conclude that device operation is similar using the figures as they are, but equal scales will be helpful.

(If these are reformatted, a formatting quirk could be corrected at the same time: the text overlaying Fig. 3a has a white background that obscures some data - unlike Fig. 3c.)

We thank the reviewer for their comments, although we do not think that replotting would improve the message of the figure significantly. We would like to note that the 'formatting quirk' they observe is not text with a white background, but in fact a section of the figure where there is no data. This is due to the fact that Fig 3a is stitched together from 3 separate data sets, whereas Fig3c is plotted from a data set that covers the full range.